# Reproductive Injury in Male Rats from Acrylamide Toxicity and Potential Protection by Earthworm Methanolic Extract

**DOI:** 10.3390/ani12131723

**Published:** 2022-07-04

**Authors:** Mohamed M. Ahmed, Amany A. Hammad, Sahar H. Orabi, Hamed T. Elbaz, Ahmed E. Elweza, Enas A. Tahoun, Mona M. Elseehy, Ahmed M. El-Shehawi, Ahmed A. Mousa

**Affiliations:** 1Department of Biochemistry and Chemistry of Nutrition, Faculty of Veterinary Medicine, University of Sadat City, Sadat City 32897, Menoufia, Egypt; mohamed.mohamed@vet.usc.edu.eg (M.M.A.); fatmasaid20@yahoo.com (A.A.H.); saher977@yahoo.com (S.H.O.); 2Department of Theriogenology, Faculty of Veterinary Medicine, University of Sadat City, Sadat City 32897, Menoufia, Egypt; hamed.elbaz@vet.usc.edu.eg (H.T.E.); ahmed.elweza@vet.usc.edu.eg (A.E.E.); 3Department of Pathology, Faculty of Veterinary Medicine, University of Sadat City, Sadat City 32897, Menoufia, Egypt; enas.tahoon@vet.usc.edu.eg; 4Department of Genetics, Faculty of Agriculture, University of Alexandria, Alexandria 21545, Alexandria, Egypt; monaahmedma@yahoo.com; 5Department of Biotechnology, College of Science, Taif University, P.O. Box 11099, Taif 21944, Saudi Arabia; a.elshehawi@tu.edu.sa

**Keywords:** acrylamide, earthworm, rats, reproductive toxicity

## Abstract

**Simple Summary:**

Acrylamide (ACR) is a vital manufacturing chemical used in polymer and copolymer production. ACR has neurotoxic, genotoxic, and carcinogenic effects. ACR monomer causes reproductive toxicity in males and diminished function by reducing sperm concentration and increasing the sperm deformity index. Earthworms have many therapeutic indications including antihypertensive, anti-allergic, anti-ulcer activities, anti-inflammatory, antitumor, antibacterial, and anti-oxidative effects. Our study showed that earthworms in a dose of 300 mg/kg prevented ACR-induced reproductive toxicity through increasing sperm motility and viability with the enhancement of sperm count, decreasing oxidative stress, and normalizing expression of p53 and Ki-67. In summary, supplementation of EE at the level of 300 mg/kg showed a protective potential effect against testicular toxicity induced by ACR.

**Abstract:**

This study examined the protective effect of earthworm extract (EE) on acrylamide (ACR)-induced reproductive dysfunction. Forty male rats were allocated into four groups *(n* = 10). The G I (control) group received distilled water (D.W.). The G II group received ACR (5 mg kg^−1^ B.W. in D.W.) 5 days per week, orally, for 3 weeks. The G III group was administered EE (300 mg kg^−1^ B.W in D.W.) 5 days per week, orally, for 3 weeks. The G IV group was pretreated with EE for 3 weeks and then co-treated with EE and ACR for an additional 3 weeks. ACR decreased the number of sperm, sperm viability, and total motility. However, it increased testosterone levels with no effect on the FSH or LH levels. Moreover, ACR increased the concentrations of malondialdehyde (MDA) and nitric oxide (NO). Meanwhile, it decreased the glutathione (GSH) concentration in testicular tissues. Notably, the expression levels of p53 and Ki-67 were increased in the degenerated spermatogenic cells and in the hyperplastic Leydig cells of the testis of the ACR-treated group, respectively. Acrylamide induced alterations in the testicular tissue architecture. Interestingly, EE restored the sperm parameters and recovered the testicular histological structures and the biochemical alterations induced by ACR. In conclusion, earthworm extract ameliorated ACR-induced reproductive toxicity via restoring the testicular antioxidant balance and suppressing p53 and Ki-67 expressions in testicular tissues.

## 1. Introduction

Acrylamide (ACR) is a crystalline, odorless, colorless monomer with high chemical activity. It is transported by passive diffusion throughout the body due to its hydrophilic capacity [1]. ACR is a vital manufacturing chemical used in polymer and copolymer production. It is created in cooked food at >120 °C in procedures such as roasting, frying, cooking, and baking of foods with a high carbohydrate ratio, which are considered a major route of ACR toxicity to humans [2]. ACR arises between asparagine amino acid and reducing sugars via the Maillard reaction during preparation of carbohydrate-rich foods at high temperatures [3]. The exposure to ACR can occur through varying routes, including inhaling ACR-contaminated air and direct contact with the toxic substance [4].

ACR has neurotoxic, genotoxic, and carcinogenic effects [5]. The ACR monomer causes reproductive toxicity in males and tumors in the endocrine system in rodents [6]. Toxicity induced by ACR can be mediated through the oxidative damage of DNA, proteins, and cells [7]. ACR induced testicular toxicity [3] and diminished testicular function by reducing the sperm concentration and increasing the sperm deformity index [8]. Several studies have shown that ACR disrupts reproductive functions [9]. ACR toxicity induced apoptosis, giant formation, and atrophy of seminiferous tubules [10].

The shift toward the usage of natural medicinal products rather than artificial products is increasing globally [11]. In this regard, substances with antioxidant properties have been given attention as probable therapeutic agents [11]. Moreover, natural herbal products protect against toxic side effects and are used as chemotherapeutic agents due to their antioxidant activity [12]. Earthworms are considered a herbal medicine [13]. The earthworm’s origin is soil, and it has a dense nutritional content [14]. Earthworms play an important role in nourishing soil fertility [15]. Earthworms are used for feeding livestock and fish and have been used widely as fishing bait [16]. Earthworms have many therapeutic indications worldwide. These indications include anti-inflammatory, antitumor, antibacterial, and anti-oxidative effects [17]. Previous studies have shown that earthworms exhibit antipyretic, antispasmodic, diuretic, antihypertensive, anti-allergic, and anti-ulcer activities [18]. Therefore, this study was designed to evaluate for the first time the protective potential of earthworm extract against testicular toxicity induced by ACR.

## 2. Materials and Methods

### 2.1. Chemicals

Acrylamide (ACR) was purchased in the form of a crystalline solid (L03670) from El-Gomhouria Company^®^, Cairo, Egypt. All the chemicals used were analytical grade.

### 2.2. Preparation of Earthworm Extracts (EEs)

A total of 3 kg of earthworms was obtained from El-Bagour, Menoufia, Egypt. Earthworm extracts were prepared following [13]. Briefly, the cleaned earthworms were killed, sliced with scissors, left to dry in the shade, and then powdered. The obtained powder (500 g) was soaked in 80% methanol at room temperature in the dark for 72 h with periodic shaking. Then, the contents were filtered using Whatman Filter Paper. The filtrate was evaporated until a soft mass was obtained. The extracts were thoroughly air dried to remove all traces of the solvent. The obtained extract was weighed (30 g) with a net crude extract of 6.6%.

### 2.3. Animals

A total of 40 male albino rats, aged four weeks and weighing 120 to 150 g, were obtained from the Al-Zyade experimental animal center (Giza, Egypt) and used for this study. The animals were kept in cages at the animal house of the Faculty of Veterinary Medicine, University of Sadat City, Egypt. The animals were kept at 22 °C ± 2, with good ventilation, and a 12 h light/dark cycle. A balanced ration (Al Wadi Company^®^, Giza, Egypt) and clean water were supplied to the rats ad libitum. The animals were kept for one week to acclimatize before the onset of the experiment. Experimental design and all procedures used were approved by the Research Ethics Committee of the Faculty of Veterinary Medicine, University of Sadat City, Egypt (VUSC-015-1-18).

### 2.4. Experimental Design

Forty rats were assigned into 4 groups (*n* = 10) as follows:

**G I**: in the control group (**C**), the rats orally received 0.5 mL distilled water (vehicle for the earthworm extract and for the ACR) 5 times per week, for 6 weeks.

**G II**: the rats were orally gavaged ACR (5 mg kg^−1^ B.W., dissolved in D.W.), following [19], 5 times per week, for 3 weeks from the 4th to the 6th week of the experiment.

**G III**: rats orally received 0.5 mL distilled water in the first three weeks of the study then received EE (300 mg/kg B.W, dissolved in D.W.) according to [20] 5 times per week for three weeks from the 4th week to 6th.

**G IV**: the rats were pretreated with EE as in group III, for 3 weeks from the 1st to the 3rd week of the experiment; then, they were co-treated with EE together with ACR as in group II for an additional 3 weeks from the 4th to the 6th week of the experiment.

### 2.5. Sampling

At the end of the experimental period, the rats fasted for 12 h and were anesthetized. Blood samples were aspirated from the orbital sinus using capillary tubes. The blood samples were left to clot. Then, the serum samples were prepared and stored at −20 °C until used for the biochemical and hormonal assays. After euthanasia, the testes were harvested, washed with saline, and divided into 2 portions; one was kept at −80 °C and used to determine the antioxidant defense system (GSH, NO) and lipid peroxidation (MDA). The other portion was maintained in neutral buffered formalin (10%) and used for histopathological and immunohistochemical investigations.

### 2.6. Evaluation of Sperm Parameters

After the experiment had ended, the rats were euthanized, and the sperm was collected from the tail of the epididymis. The cauda epididymis was dissected and placed in 1 mL phosphate buffer saline (PBS) that was pre-warmed. The total sperm count was estimated with a Neubauer chamber in a drop of the sperm suspension. Smooth agitation was applied along with tissue tearing to make the spermatozoa swim out into the PBS, and the sperm suspension was placed in the Eppendorf tube [21]. The samples were incubated at 37 °C, 5% CO_2_, for 20 min, for further analysis of the sperm.

### 2.7. Evaluating Sperm Concentration and Total Motility

The sperm total motility and concentration were determined as described by [22]. Briefly, the percentages of motile sperm were determined by a phase-contrast microscope, heated stage, under high-power magnification (40×). The motility percentage of sperm was graded as proportion of progressive and non-progressive spermatozoa. Several microscopic fields were examined to determine sperm motility percentage. The total sperm number was estimated by a Neubauer chamber in a drop of the resulting suspension of sperm [22].

### 2.8. Evaluation of Sperm Viability

On a microscope slide, equal volumes of sperm sample and eosin stain (0.5%) were mixed, covered with a coverslip, and examined. Two hundred sperm were counted [21]. Light microscopy at 100× was used to determine live and dead spermatozoa. The live sperm appeared unstained, while the dead sperm were stained red.

### 2.9. Assaying of Reproduction-Related Hormones

In vitro assays were used for the quantitative determination of serum luteinizing hormone (LH) (LIAISON LH, REF: 312201), follicle-stimulating hormone (FSH) in serum (LIAISON^®^ FSH. Code 312251), and testosterone hormones (LIAISON^®^Testosterone REF: 310410); the tests were performed by ELISA.

### 2.10. Assaying of Oxidative Stress Biomarkers in Testicular Tissues

Specific commercial kits for assaying antioxidant biomarkers (GSH, CAT. No. GT 2519) and lipid peroxidation biomarkers (MDA, CAT. No. MD 2529; NO, CAT. No. GT 2533) were obtained from Biodiagnostic (Giza, Egypt). The manufacturer’s instructions were followed to perform all measurements.

### 2.11. Histopathological Examination

After necropsy, the testes specimens were immediately fixed in neutral buffered formalin (10%). The specimens were subjected to dehydration in ascending ethanol concentrations. Then, they were cleared in xylene and embedded in paraffin wax. Four-micrometer-thick sections were prepared and stained with hematoxylin and eosin [23].

### 2.12. Immunohistochemical Investigations

Both p53 and Ki-67 were detected immunohistochemically in the testicular sections, according to [24]. The slides were incubated with primary antibodies (Thermo Fisher Scientific, Waltham, MA, USA) against Ki-67 (polyclonal antibody, CAT. NO. RM-9105-R7) and p53 (monoclonal antibody, CAT. NO. RB-9043-R7) at a concentration of 1μg/mL for 2 h at room temperature. The slides were rinsed with PBS four times. Incubation with the corresponding secondary antibody was conducted for 30 min at room temperature. Then, the sections were exposed to avidin and biotinylated HRP complex for 30 min. The slides were incubated with peroxidase-compatible chromogen for 10 min, washed 4 times with DW, and finally counterstained with hematoxylin to stain the nuclei. **Quantitative analysis of immunohistochemical parameters:** After performing the color deconvolution command, the number of p53 and Ki 67- immune positive cells in the seminiferous tubules was counted under a light microscope using an image-analyzing system equipped with a computer-based CCD camera (software: Optimas 6.5, Cyber Metrics, Scottsdale, AZ, USA) and compared between groups. Cell counts were obtained by averaging the counts from randomly chosen fifteen microscopic high-power fields (15 HPF) at ×40 (3 tissue sections per group and 5 fields per tissue section).

### 2.13. Statistical Analysis

Data are presented as means ± standard errors. Statistical significances were determined by one-way ANOVA. Statistical analysis was conducted using Version 16 of SPSS. A *p* value less than 0.05 was regarded as statistically significant.

## 3. Results

### 3.1. Effect of ACR and EE Extract on Sperm Count, Motility, and Viability

The administration of ACR reduced the sperm count, sperm total motility, and viability, while EE alone (GII) showed maximum sperm motility and viability with an enhancement of the sperm count. Additionally, rats co-treated with EE and ACR regained normal sperm count, total motility, and viability (Table 1).

### 3.2. Effect of ACR and EE Extract on Testicular Oxidative Stress and Antioxidant Biomarkers

Acrylamide increased the testicular MDA and NO contents and decreased the GSH level (Table 2). In rats pretreated with EE and then co-treated with EE and ACR, testicular MDA was decreased to the normal control level, while testicular GSH was significantly increased with no change in NO level. The earthworm extract alone significantly decreased NO levels but did not affect testicular MDA or GSH (Table 2).

### 3.3. Effect of ACR and EE Extract on Hormonal (FSH, LH, Testosterone) Blood Concentrations

Serum hormonal analysis revealed that the administration of ACR (G II) increased the testosterone hormone with no effect on the FSH and LH concentrations (Table 3). Pre-treatment with EE, then co-treatment with EE and ACR (G IV), caused a reduction in serum testosterone level compared to the ACR treatment of the animals (GII). There was no significant difference in serum testosterone level between the G III, administered EE alone, and the control group (GI).

### 3.4. Effect of ACR and EE Extract on Histology of Testicular Tissues

In ACR-treated rats (G II), seminiferous tubules showed loss of spermatogenic epithelium, desquamation of spermatogenic epithelium in their lumen, vacuolation in the germinal layers, and Leydig cell hyperplasia (Figure 1B). The EE-treated rats (G III) showed normal histologic architectures of seminiferous tubules and interstitial tissues, as in the control group (Figure 1C). The rats in the G IV group showed nearly normal seminiferous tubules with congestion of interstitial blood vessels (Figure 1D).

### 3.5. Effect of ACR and EE Extract on Expression of p53 and Ki- 67 in Rat Testicular Tissues

Immunohistochemical examination of the testicular tissues for p53 expression demonstrated that the ACR-treated rats (G II) showed a positive affinity for p53 in the degenerated spermatogenic cells (Figure 2B) when compared to the control group (Figure 2A). The EE-treated rats (G III) showed no expression, as in the control group (Figure 2C). The rats in the G IV group showed a mild positive affinity for p53 in some spermatogenic cells (Figure 2D). In contrast to p53 expression, the expression of Ki-67 was increased in the spermatogenic cells of seminiferous tubules compared to that of the control group. There was also strong positive Ki-67 immunostaining in spermatogonia and spermatocytes in this group (Figure 3A). On the contrary, immunostaining of Ki-67 in the ACR-treated rats (G II) was negative in the seminiferous tubules. However, there was strong Ki-67 positive immunostaining in the hyperplastic Leydig cells (Figure 3B). Strong Ki-67 positive immunostaining was shown in the spermatogenic cells of the seminiferous tubules from the EE-treated group (GIII) (Figure 3C). The rats in the G IV group showed a partial restoring of Ki 67 expression in the spermatogenic cells of the seminiferous tubules (Figure 3D). Immunohistochemical quantitative analysis in our current experiment revealed that acrylamide induced a significant elevation of the numbers of p53-positive cells compared to control rats, and this increase was significantly down-regulated by EE pre-treatment. On the contrary, the numbers of Ki-67-positive cells were significantly diminished by acrylamide treatment, and this decrease was significantly up-regulated by EE pre-treatment which confirmed that earthworm reversed the inhibition of adult rat germ cell proliferation induced by acrylamide as presented in Table 4.

## 4. Discussion

Acrylamide is one of the main environmental health problems [25]. Various animal studies have suggested that ACR harms male reproductive function [9]. Moreover, [10] showed that ACR treatment resulted in seminiferous tubule atrophy, reduced sperm viability, and the development of abnormal spermatozoa in rats. Earthworm extract is extensively used in traditional medicine in China [14].

The testis is the target organ of action for ACR [26]. In the current study, ACR severely disrupted reproductive parameters in rats, causing a sharp decrease in sperm count, sperm total motility, and sperm viability. Similar data were obtained by [6,27]. Moreover, in this study, ACR treatment stimulated testosterone production by Leydig cells. This might be attributed to Leydig cell hyperplasia as confirmed by the histopathological findings. Parallel to our findings, it was demonstrated that ACR causes a significant elevation in serum testosterone with hyperplasia of Leydig cells in rats [19]. Conversely, [28] showed that ACR decreased serum testosterone, FSH, and LH hormones. They ascribed these results to oxidative stress. Moreover, [27] reported that ACR decreased the testosterone level and attributed this finding to the death of Leydig cells. The harmful effect of ACR on sperm parameters and testicular tissues in the current study might be explained by the oxidative stress resulting from ACR administration. These results are in line with those of [29]. Sperm count, motility, and viability were all improved by the EE. Treatment with EE reduced MDA levels while increasing GSH levels in testicular tissue, possibly due to its antioxidant effects. These results agree with [30], who showed that EE restored liver MDA and the GSH changes caused by SiNP administration. Furthermore, the current results match those of [31], who stated that EE increased GSH, which was diminished by CCL_4_ administration. They attributed their results to the free radical scavenging activity of EE.

Parallel to these findings, the histopathological examination revealed that ACR caused obvious alterations in the testicular tissue. These alterations could be attributed to ACR-induced oxidant/antioxidant imbalance. These results agree with the results presented by [19]. Leydig cell hyperplasia and loss of germ cells have been also reported in mice [32], which could explain the rise in serum testosterone observed in the GII group in response to the ACR treatment. Partial loss of germ cells and Sertoli cells in the epithelium of the seminiferous tubules of ACR-treated rats suggests ACR interference with rat testicular functions [19]. On the other hand, histological changes were greatly improved with the EE treatment due to its antioxidant effect. Supporting our explanation, it was shown that earthworm paste has anti-oxidative and anti-ulcer properties [17].

p53 is a proapoptotic marker and tumor suppressor gene. It plays an essential role in tumor cell sensitivity to apoptosis [33]. On the other hand, Ki-67 expression is a cell proliferation marker [34]. The results show clear up-regulation of p53 expression in the spermatogenic cells from the ACR-treated group. However, Ki-67 expression was up-regulated in the testicular Leydig cells from the ACR-treated group. Meanwhile, it was down-regulated in the spermatogenic cells of the seminiferous tubules. This could explain the decreased number of sperm and increased serum testosterone concentration in the ACR-treated group (GII). The co-administration of EE and ACR counteracted these effects. The results of previous studies agree with these findings [35,36]. Earthworm extract increases Ki-67 expression in the skin and accelerates cell proliferation and the differentiation of fibroblasts in the wound healing of mice [36]. In this study, ACR caused oxidative stress and apoptosis in rat testes. Treatment with EE ameliorated the histopathological alterations observed in the ACR-treated rats. Moreover, EE treatment decreased the expression of p53 in the spermatogenic cells and Ki-67 expression in the Leydig cells. This finding could explain the effect of EE on ACR-induced histopathological alterations in rat testes. The limitation of the study was the fact that the semen was evaluated subjectively and with not innovative methods such as fluorescent dyes or sperm class analyzer.

## 5. Conclusions

In conclusion, this study showed that ACR causes reproductive toxicity via the induction of oxidative stress that alters the expression of p53 and Ki-67 and, subsequently, induces testicular damage. In contrast, EE prevented ACR-induced reproductive toxicity through decreasing the oxidative stress and normalizing the expression of p53 and Ki-67. Our results indicate that EE is a protective candidate against ACR-induced reproductive toxicity.

## Figures and Tables

**Figure 1 animals-12-01723-f001:**
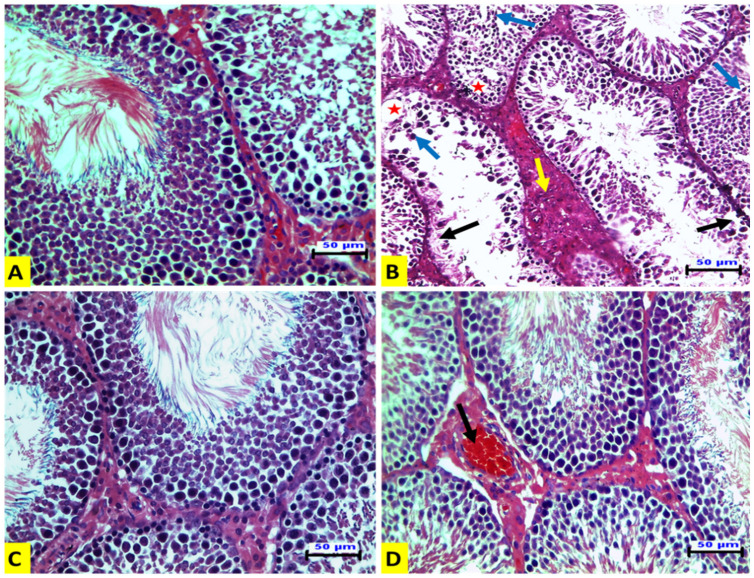
**Representative histopathological photomicrographs of transverse sections of rat testes** (hematoxylin and eosin stain X_20_, scale bar 50 μm). (**A**) The control group has normal histologic structures of the seminiferous tubules and interstitial tissue. (**B**) The ACR-treated group has seminiferous tubules with a loss of spermatogenic epithelium leaving a denuded basement membrane (black arrow). There is also desquamated spermatogenic epithelium (blue arrow). Vacuolation in germinal layers (star) and hyperplasia of Leydig cells (yellow arrow) are also shown. (**C**) The EE-treated group shows normal histologic structures of the seminiferous tubules and interstitial tissue. (**D**) The EE + ACR group shows nearly normal seminiferous tubules and congestion of interstitial blood vessels (arrow).

**Figure 2 animals-12-01723-f002:**
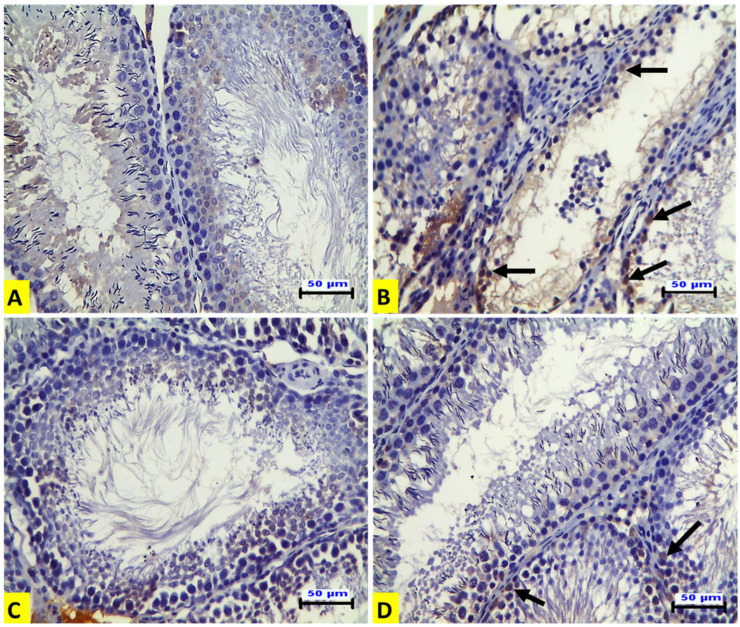
**Representative histopathological photomicrographs of transverse sections from rat testes** (IHC for p53 X_20_, scale bar 50 μm). (**A**) The control group shows a negative immune reaction for p53. (**B**) The ACR-treated group shows positive affinity for p53 in the degenerated spermatogenic cells (arrows). (**C**) The EE-treated group shows no immune expression. (**D**) The EE + ACR group shows mild positive affinity for p53 in some spermatogenic cells (arrows).

**Figure 3 animals-12-01723-f003:**
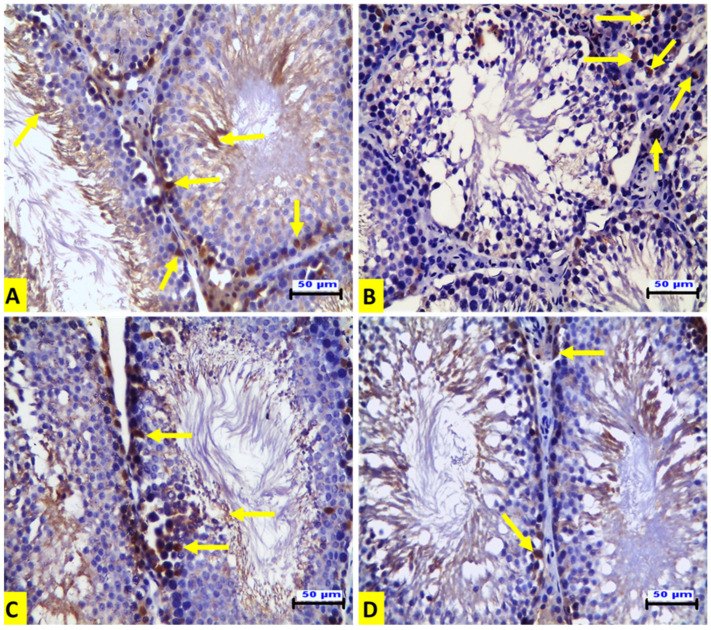
**Representative histopathological photomicrographs of transverse sections from rat testes** (IHC for Ki-67 X_20_, scale bar 50 μm). (**A**) The control group exhibits Ki-67 immunostaining in the spermatogenic cells of the seminiferous tubules with strong positive Ki-67 immunostaining in the spermatogonia and spermatocytes (arrows). (**B**) The ACR-treated group shows negative Ki-67 immunostaining in the seminiferous tubules and strong Ki-67-positive immunostaining in the hyperplastic Leydig cells. (**C**) The EE-treated rats show strong Ki-67-positive immunostaining in the spermatogenic cells of the seminiferous tubules. (**D**) The EE + ACR group shows weak immunostaining for Ki-67 in the spermatogenic cells of the seminiferous tubules.

**Table 1 animals-12-01723-t001:** Effect of acrylamide and/or earthworm extract on sperm count, motility, and viability.

	GI	GII	GIII	GIV
**Count (×10** ** ^6^ ** **)**	28.25 ± 1.77 ^a^	14.83 ± 1.07 ^b^	50 ± 2.04 ^c^	26.75 ± 0.25 ^a^
**Rapid motility (%) (Grade a)**	26 ± 1.20 ^a^	8.75 ± 0.72 ^b^	47.1 ± 4.62 ^c^	16.42 ± 1.99 ^b^
**Slow motility (%) (Grade b)**	38 ± 1.64 ^a^	16.25 ± 1.38 ^b^	30 ± 2.04 ^c^	31.87 ± 2.3 ^a, c^
**Non-progressive motility (%) (Grade c)**	16.42 ± 1.09 ^a^	33.75 ± 1.38 ^b^	11.42 ± 1.09 ^a^	31.42 ± 2.15 ^b^
**Immotile sperm (%) (Grade d)**	19.6 ± 0.25 ^a^	38.75 ± 0.72 ^b^	11 ± 1.20 ^c^	21 ± 1.88 ^a^
**Total motility (%) (Grade a, b, c)**	80.42 ± 3.93 ^a^	58.75 ± 3.48 ^b^	88.52 ± 7.75 ^c^	79.71 ± 6.44 ^a^
**Viability (%)**	81.33 ± 1.1 ^a^	60.55 ± 1.68 ^b^	85.45 ± 1.5 ^a^	76.42 ± 1.09 ^a, c^

Data are presented as means ± SEM. A *p* value ˂ 0.05 was regarded as statistically significant. Values with different letters in the same row differ significantly between groups. **GI** (control) was gavaged with distilled water (D.W.). **GII** was administered acrylamide (ACR; 5 mg kg^−1^ B.W. in D.W.) orally for 3 weeks. **GIII** was given EE (300 mg kg^−1^ B.W in D.W.) orally for 3 weeks. **GIV** was pretreated with EE for 3 weeks, then co-treated with ACR for another 3 weeks.

**Table 2 animals-12-01723-t002:** Effect of acrylamide and/or earthworm extract on testicular oxidative stress and antioxidant biomarkers.

	GI	GII	GIII	GIV
**GSH** **(mmol/g. tissue)**	9.46 ± 0.48 ^a, c^	6.49 ± 0.58 ^b^	10.05 ± 0.53 ^a^	8.14 ± 0.22 ^c^
**NO (mmol/g. tissue)**	219.77 ± 11.37 ^c^	284.77 ± 7.10 ^a^	165.52 ± 17.56 ^b^	265.92 ± 11.17 ^a^
**MDA (nmol/g. tissue)**	47.05 ± 1.74 ^b^	58.12 ± 2.01 ^a^	51.23 ± 1.62 ^b^	50.90 ± 1.33 ^b^

Data are presented as means ± SEM. A *p* value ˂ 0.05 was regarded as statistically significant. Values with different letters in the same row differ significantly between groups. **GI** (control) was gavaged with distilled water (D.W.). **GII** was administered acrylamide (ACR; 5 mg kg^−1^ B.W. in D.W.) orally for 3 weeks. **GIII** was given EE (300 mg kg^−1^ B.W in D.W.) orally for 3 weeks. **GIV** was pretreated with EE for 3 weeks and then co-treated by EE and ACR for an additional 3 weeks. GSH, reduced glutathione; NO, nitric oxide; MDA, malondialdehyde; SEM, standard error of the mean.

**Table 3 animals-12-01723-t003:** Effect of acrylamide and/or earthworm extract on serum gonadal and pituitary hormones.

	GI	GII	GIII	GIV
**Testosterone (ng/dL)**	3.96 ± 0.17 ^c^	8.32 ± 0.19 ^a^	4.82 ± 0.09 ^b^	3.80 ± 0.13 ^c^
**FSH (mIU/mL)**	2.33 ± 0.12	2.37 ± 0.11	2.49 ± 0.1	2.34 ± 0.13
**LH (mIU/mL)**	1.12 ± 0.04	1.05 ± 0.06	1.21 ± 0.08	1.24 ± 0.05

Data are presented as means ± SEM. A *p* value ˂ 0.05 was regarded as statistically significant. Values with different letters in the same row differ significantly between groups. **GI** (control) was gavaged with distilled water (D.W.). **GII** was administered acrylamide (ACR; 5 mg kg^−1^ B.W. in D.W.) orally for 3 weeks. **GIII** was given EE (300 mg kg^−1^ B.W in D.W.) orally for 3 weeks. **GIV** was pretreated with EE for 3 weeks and co-treated by EE and ACR for an additional 3 weeks. FSH, follicle-stimulating hormone. LH, luteinizing hormone. SEM, standard error of the mean.

**Table 4 animals-12-01723-t004:** Quantitative analysis of immunohistochemical parameters P 53 and Ki-67 in testicular tissue.

Parameters	GI	GII	GIII	GIV
P 53	0.6 ± 0.2 ^c^	14.7 ± 0.97 ^a^	0.2 ± 0.01 ^c^	5.87 ± 0.7 ^b^
Ki-67	25.4 ± 0.9 ^a^	0.73 ± 0.o2 ^c^	26 ± 0.9 ^a^	13.7 ± 0.9 ^b^

Data are presented as means ± SEM (standard error of means) *n* = 15. A *p* value ˂ 0.05 was regarded as statistically significant. Values with different letters in the same row differ significantly between groups. GI (control): was gavaged with distilled water (D.W.). GII: was administered acrylamide (ACR; 5 mg kg^−1^ B.W in D.W.) orally for 3 weeks. GIII: was given EE (300 mg kg^−1^ B.W in D.W.) orally for 3 weeks. GIV: was pretreated with EE for 3 weeks, then co-treated by EE and ACR for additional 3 weeks. P 53: protein 53; Ki-67: Ki-67 protein.

## Data Availability

Data supporting reported results can be found at the article website.

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
