# Peer review of "Reproductive Injury in Male Rats from Acrylamide Toxicity and Potential Protection by Earthworm Methanolic Extract"

_animals, 2022, doi:10.3390/ani12131723_

Round 1

Reviewer 1 Report

Authors should be resubmitted after comprehensive proofreading according to journal policy. This draft did not contains simple summary and not followed journal policy. Therefore, this research should be resubmitted after modifications. Additionally, this draft does not include information regarding Ethics Committee or Institutional Review Board approval.

Author Response

Reviewer: 1#

Comment 1:

Authors should be resubmitted after comprehensive proofreading according to journal policy. This draft did not contain simple summary and not followed journal policy. Therefore, this research should be resubmitted after modifications.

Response:

The manuscript has undergone English language editing by MDPI editing service. The text has been checked for correct use of grammar and common technical terms and edited to a level suitable for reporting research in a scholarly journal. Also, the manuscript was modified according to the journal policy, and we added a simple summary ahead of the abstract as follows.

Simple summary

Acrylamide (ACR) is a vital manufacturing chemical used in polymers and copolymers production. ACR has neurotoxic, genotoxic, and carcinogenic effects. ACR monomer causes reproductive toxicity in males and diminished its functions by reducing sperm concentration and increasing sperm deformity index. Earthworms have many therapeutic indications including antihypertensive, anti-allergic, anti-ulcer activities, anti-inflammatory, antitumor, antibacterial, and anti-oxidative effects. Our study showed that Earthworm methanolic extract (EE) with a dose of 300 mg/kg prevented ACR-induced reproductive toxicity through increasing sperm motility and viability with the enhancement of sperm count, decreasing oxidative stress, and normalizing expression of p53 and Ki 67. The present results clearly suggest the EE as a promising protective and therapeutic candidate against testicular toxicity induced by ACR.

Comment 2:

Additionally, this draft does not include information regarding Ethics Committee or Institutional Review Board approval

We have added information regarding Ethics Committee or Institutional Review Board approval within manuscript at section 2.3. Animals as follows: Experimental   design   and   all   procedures   used were   approved by the Research Ethics Committee of the Faculty of Veterinary Medicine, University of Sadat City, Egypt (VUSC-015-1-18(.

Thank you for your time and effort.

Reviewer 2 Report

As one of the main environmental health problems, Acrylamide (ACR) may harm the male reproductive function. Earthworm is a herbal medicine which has versatile functions. However, whether the earthworm extraction (EE) can reverse the ACR-induced reproductive damage is still unclear. This study tackled this question by using rat model. The experimental design is reasonable. The authors used appropriate methods and obtained the data, which are solid and can support the conclusions. I suggest the authors think about the following issues to improve their manuscript.

  1. In 2.3 Animals, it's should be stated whether this study follows the guidence of the related  Ethics and welfare of laboratory animals.
  2. For Figure 2 and Figure 3, it's better to do the quantitative analysis of p53 positive cells and ki-67 positive cells among four groups.

Author Response

Reviewer: 2#

Comment 1:

In 2.3 Animals, it's should be stated whether this study follows the guidance of the related  Ethics and welfare of laboratory animals.

Response:

We have added information regarding Ethics Committee or Institutional Review Board approval within manuscript at section 2.3. Animals as follows: Experimental   design   and   all   procedures   used were   approved by the Research Ethics Committee of the Faculty of Veterinary Medicine, University of Sadat City, Egypt (VUSC-015-1-18).

Comment 2:

  For Figure 2 and Figure 3, it's better to do the quantitative analysis of p53 positive cells and ki-67 positive cells among four groups

Response:

This comment was addressed as recommended by the reviewer at the methodology section.2.11 as follows, Immunohistochemical investigations:

Quantitative analysis of immunohistochemical parameters: After performing color deconvolution command, the number of p53 and Ki 67- immune positive cells in the seminiferous tubules were counted under light microscope using an image analyzing system equipped with a computer-based CCD camera (software: Optimas 6.5, Cyber Metrics, Scottsdale, AZ, U.S.A.) and were compared between groups. Cell counts were obtained by averaging the counts from randomly chosen fifteen microscopic high-power fields (15 HPF) at X40 (3 tissue sections per group and 5 fields per tissue section).

Also, we have added the following at the results section 3.5. Earthworm extract ameliorated ACR-induced expression of p53 and Ki- 67 in rat testicular tissues:

Immunohistochemical quantitative analysis of our current experiment revealed that acrylamide induced a significant elevation of numbers of p53-positive cells compared to control rats and this increase was significantly down-regulated by EE pre-treatment. In contrary, numbers of Ki67-positive cells were significantly diminished by acrylamide treatment and this decrease was significantly up-regulated by EE pre-treatment which confirmed that earth worm reversed the inhibition of adult rat germ cell proliferation induced by acrylamide as presented in table (4).

Moreover, we have added Table 4 at the end of the results section 3.5.

Table (4): Quantitative analysis of immunohistochemical parameters P 53andKi-67in testicular tissue:

Parameters

GI

GII

GIII

GIV

P 53

0.6±0.2c

14.7±0.97a

0.2±0.01 c

5.87±0.7b

Ki-67

25.4±0.9 a

0.73±0.o2 c

26±0.9 a

13.7±0.9 b

Data are presented as means ± SEM (standard error of means) n = 15. a p value ˂ 0.05 was regarded as statistically significant. Values with different letters at the same row differ significantly.G І (control): was gavaged with distilled water (D.W). G ІІ: wasadministered acrylamide (ACR; 5 mg kg-1 B.W in D.W) orally for 3 weeks. G ІІІ: was given EE (300 mg kg-1 B.W in D.W) orally for 3 weeks. G ІV: was pretreated with EE for 3 weeks, then co-treated by EE and ACR for additional 3 weeks. P 53: protein 53; Ki-67: Ki-67 protein.

Thank you for your time and effort.

Reviewer 3 Report

The article is not suitable for publication in its current form. It is required to supplement the data with sperm quality results. Assessment of sperm motility and sperm viability (eosin stain) is not enough for checking sperm quality parameters.

The stylistics and punctuation errors must be removed.

Author Response

Reviewer: 3#

Comment 1:

The article is not suitable for publication in its current form. It is required to supplement the data with sperm quality results. Assessment of sperm motility and sperm viability (eosin stain) is not enough for checking sperm quality parameters.

Response:

Thank you for such professional comment. Although we used traditional protocols for assessing sperm quality parameters, these methods are approved as a representative indicator for sperm quality parameters in well-reputed international journals. Moreover, our data are supported by antioxidant biomarkers assays that showed a significant increase in the oxidative stress biomarkers as well as histological investigations that showed clear histopathological alterations in the acrylamide-intoxicated rats. Please find our recent article “Mai M. Eissa | Mohamed M. Ahmed | Mabrouk A. Abd Eldaim | Sahar H. Orabi | Hamed T. Elbaz | Mostafa A. Mohamed | Ahmed E. Elweza | Ahmed A. Mousa. Methanolic extract of Chlorella vulgaris protects against sodium nitrite-induced reproductive toxicity in male rats. Andrologia. 2020;00: e13811. https://doi.org/10.1111/and.13811 “

Comment 2:

The stylistics and punctuation errors must be removed.

 Response:

 All stylistics and punctuation errors were removed in the revised manuscript. The manuscript has undergone English language editing by MDPI. The text has been checked for correct use of grammar and common technical terms and edited to a level suitable for reporting research in a scholarly journal.

Thank you for your time and effort

Reviewer 4 Report

The study provides new useful information on the protective effect of Earthworm on rat spermatozoa. The manuscript presents a well-performed experimental design, making different tests to understand the mechanism underlying the effects of Eartworm.

However, However, an English revision is mandatory and are some major issues that should be improved for the final approval.

Please check in all the Manuscript the punctuation errors.

Abstract

Lines 20-21: Please add “;” between groups.

Lines 20-27: Please reword the results, they are not understandable. It may be better to list the results of each group, for example “In group II there was a decrease of total motility and ….” or replace with “ACR decreased number of sperms, sperm viability and total motility although increased testosterone levels with no effect on FSH or LH levels. ACR increased concentrations of malondialdehyde (MDA) and nitric oxide (NO). EE decreased reduced glutathione 24 (GSH) concentration in testicular tissues. Notably, expression levels of p53 and Ki-67 were increased 25 in the degenerated spermatogenic cells and in the hyperplastic Leydig cells of the testis of ACR- treated group respectively”, if is a correct interpretation of your results.  Please avoid or pay attention of the use of “However/ Meanwhile/Furthermore”.

Materials and methods

Lines 92-93: Please specify if group III received 0.5 mL distilled water in the first three weeks of the study.

Please delete the dots and colons at the end of the title of each paragraph.

Results

It is better to change the name of the paragraphs in “Effect of ACR and EE extract on sperm count, motility, and viability / on testicular oxidative stress and antioxidant biomarkers/ hormonal (FSH, LH, Testosterone) blood concentrations/ histology of testicular tissues / expression of p53 and Ki- 67 in rat testicular tissues”.

In the Tables you should specify that the “Values with different letters at the same row differ significantly between groups”. Moreover, the value of the first column (Group II) should be in every table “a” and not “c” as in Table 2 and 3, please correct them.

Line 172: In the Table 1 the caption above is incomplete.

Discussion

Line 253: Please replace “Population ACR exposure is one of the main environmental health problems” with Reproductive toxicity following acrylamide exposure is one of the main environmental health problems” or rephrase it.

Lines 256: Please add the use in traditional medicine in China of the Earthworm extract.

Lines 258-260: Please replace with “In the current study, ACR severely disrupted reproductive parameters in rats, causing a sharp decrease in sperm count, sperm total motility and sperm viability” or “decreasing sperm count, sperm total motility and sperm viability”.

Lines 270-272: please rephrase.

Lines 271-275: please add comments on the further analysis that could be performed to verify the antioxidant effect of EE (other sperm analysis, such as DNA fragmentation, acrosome reactions..).

In this study only a subjective evaluation of the semen with old tests was performed, please add a phrase in the discussion on the limitation of this study, with something like: “The limitation of the study was the fact that the semen was evaluated subjectively and with not innovative methods.”  (not use of fluorescent dyes or sperm class analyzer).

Author Response

Response to reviewers’ comments

Dear Editor,

Thank you very much for your time and effort.

All thanks and gratitude’s to the respected reviewers whose valuable comments surely will improve our manuscript. All claims and comments were addressed, and the manuscript was amended as required. You can follow changes by using track changes facilities.

Reviewer: 4#

Comment 1:

Please check in all the Manuscript the punctuation errors.

 Response:

 The manuscript has undergone English language editing by MDPI, and all punctuation errors were removed in the revised manuscript.

Comment 2:

Lines 20-21: Please add “;” between groups.

Response:

We added “;” between groups.

Comment 3:

Lines 20-27: Please reword the results, they are not understandable. It may be better to list the results of each group, for example “In group II there was a decrease of total motility and ….” or replace with “ACR decreased number of sperms, sperm viability and total motility although increased testosterone levels with no effect on FSH or LH levels. ACR increased concentrations of malondialdehyde (MDA) and nitric oxide (NO). EE decreased reduced glutathione 24 (GSH) concentration in testicular tissues. Notably, expression levels of p53 and Ki-67 were increased 25 in the degenerated spermatogenic cells and in the hyperplastic Leydig cells of the testis of ACR- treated group respectively”, if is a correct interpretation of your results.  Please avoid or pay attention of the use of “However/ Meanwhile/Furthermore”.

Response:

The manuscript has undergone English language editing by MDPI. The text has been checked for correct use of grammar and common technical terms and edited to a level suitable for reporting research in a scholarly journal.

Comment 4:

Lines 92-93: Please specify if group III received 0.5 mL distilled water in the first three weeks of the study.

Response:

 In Line 96 we rewrote the sentence:

Rats were orally received 0.5 mL distilled water in the first three weeks of the study then received EE (300 mg/kg B.W, dissolved in D.W) according to [20] 5 times a week for three weeks starting from the 4th week to 6th.

Comment 4:

Please delete the dots and colons at the end of the title of each paragraph.

Response:

All dots and colons at the end of the title of each paragraph were deleted.

Comment 5:

It is better to change the name of the paragraphs in “Effect of ACR and EE extract on sperm count, motility, and viability / on testicular oxidative stress and antioxidant biomarkers/ hormonal (FSH, LH, Testosterone) blood concentrations/ histology of testicular tissues / expression of p53 and Ki- 67 in rat testicular tissues”.

Response:

In results section we have changed the following:

  1. 1. Earthworm extract (EE) improved sperm parameters disrupted by acrylamide (ACR) into Effect of ACR and EE extract on sperm count, motility, and viability
  2. 2. EE reduced oxidative stress induced by ACR in rat testicular tissues into Effect of ACR and EE extract on testicular oxidative stress and antioxidant biomarkers.
  3. 3. EE ameliorated ACR-induced hormonal (FSH, LH, and Testosterone) changes into Effect of ACR and EE extract on hormonal (FSH, LH, Testosterone) blood concentrations.
  4. 4. EE ameliorated the ACR-induced histopathological alterations in testicular tissues into Effect of ACR and EE extract on histology of testicular tissues.
  5. 5. Earthworm extract ameliorated ACR-induced expression of p53 and Ki- 67 in rat testicular tissues into Effect of ACR and EE extract on expression of p53 and Ki- 67 in rat testicular tissues.

Comment 6:

In the Tables you should specify that the “Values with different letters at the same row differ significantly between groups”.

Response:

We rewrite below each table sentence (Values with different letters at the same row differ significantly between groups)

Comment 7:

Line 172: In Table 1 the caption above is incomplete.

Response:

We have completed the missing sentence of table 1 caption as follows:

G ІV was pretreated with EE for 3 weeks, then cotreated with ACR for another 3 weeks.

Comment 8:

Line 253: Please replace “Population ACR exposure is one of the main environmental health problems” with Reproductive toxicity following acrylamide exposure is one of the main environmental health problems” or rephrase it.

Response:

All grammatical mistakes were corrected after the manuscript has undergone English language editing by MDPI

 Comment 9:

Lines 256: Please add the use in traditional medicine in China of the Earthworm extract.

Response:

We have previously mentioned the use of earth worm in traditional medicine at the introduction in lines 56-62: Earthworms have many therapeutic indications worldwide. These indications include anti-inflammatory, antitumor, antibacterial, and anti-oxidative effects [17]. Previous studies showed that earthworms exhibit antipyretic, antispasmodic, diuretic, antihypertensive, anti-allergic, anti-ulcer activities [18].

Comment 10:

Lines 258-260: Please replace with “In the current study, ACR severely disrupted reproductive parameters in rats, causing a sharp decrease in sperm count, sperm total motility and sperm viability” or “decreasing sperm count, sperm total motility and sperm viability”.

Response:

 In Lines 300-301: we rewrite sentence to become:

In the current study, ACR severely disrupted reproductive parameters in rats, causing a sharp decrease in sperm count, sperm total motility and sperm viability” or “decreasing sperm count, sperm total motility and sperm viability.

Comment 11:

Lines 270-272: please rephrase.

Response:

Lines 312-313: we rephrased the sentence “The EE enhanced sperm count, motility, and viability. treatment with EE decreased MDA level and   increased   GSH level in testicular tissue which could be attributed to its antioxidant properties.” to “Sperm count, motility, and viability were all improved by the EE. Treatment with EE reduced MDA levels while increasing GSH levels in testicular tissue, possibly due to its antioxidant effects.”

Comment 12:

Lines 271-275: please add comments on the further analysis that could be performed to verify the antioxidant effect of EE (other sperm analysis, such as DNA fragmentation, acrosome reactions.).

Response:

Thank you for such valuable comment. This will be carefully considered in our future study. We are planning to conduct a more detailed study through which we will fractionate the bioactive materials in EE and identify the candidate fraction responsible for theses obtained results and to elucidate the possible underlying signaling mechanism.

Comment 13:

In this study only a subjective evaluation of the semen with old tests was performed, please add a phrase in the discussion on the limitation of this study, with something like: “The limitation of the study was the fact that the semen was evaluated subjectively and with not innovative methods.”  (not use of fluorescent dyes or sperm class analyzer).

Response:

We have added “The limitation of the study was the fact that the semen was evaluated subjectively and with not innovative methods like use of fluorescent dyes or sperm class analyzer” at the end of discussion (lines 346-347). Moreover, the used protocols are approved as a representative indicator for sperm quality parameters in well-reputed international journals. In addition, our data are supported by antioxidant biomarkers assays that showed a significant increase in the oxidative stress biomarkers as well as histological investigations that showed clear histopathological alterations in the acrylamide-intoxicated rats. Please find our recent article “Mai M. Eissa | Mohamed M. Ahmed | Mabrouk A. Abd Eldaim | Sahar H. Orabi | Hamed T. Elbaz | Mostafa A. Mohamed | Ahmed E. Elweza | Ahmed A. Mousa. Methanolic extract of Chlorella vulgaris protects against sodium nitrite-induced reproductive toxicity in male rats. Andrologia. 2020;00: e13811. https://doi.org/10.1111/and.13811 “ and others Majid Pourentezari, Alireza Talebi, Abulghasem Abbasi, Mohammad Ali Khalili, Esmat Mangoli, Morteza Anvari. Effects of acrylamide on sperm parameters, chromatin quality, and the level of blood testosterone in mice. Iranian Journal of Reproductive Medicine Vol. 12. No. 5. pp: 335-342, May 2014.

Osama E, Galal AAA, Abdalla H, El Sheikh SMA. Chlorella vulgaris ameliorates testicular toxicity induced by deltamethrin in male rats via modulating oxidative stress. Andrologia. 2019;51:e13214. https://doi.org/10.1111/and.13214

Thank you for your time and effort.

Round 2

Reviewer 1 Report

It is suitable to be published with present form. 

Author Response

Thank you

Reviewer 4 Report

Thank you for the changes.  The manuscript can accepted in present form.

Author Response

Thank you